# The Impact of Covid-19 and the Effect of Psychological Factors on Training Conditions of Handball Players

**DOI:** 10.3390/ijerph17186471

**Published:** 2020-09-05

**Authors:** Daniel Mon-López, Alfonso de la Rubia Riaza, Mónica Hontoria Galán, Ignacio Refoyo Roman

**Affiliations:** Facultad de Ciencias de la Actividad Física y del Deporte (INEF - Sports Department), Universidad Politécnica de Madrid, 28040 Madrid, Spain; alfonso.delarubia@upm.es (A.d.l.R.R.); monica.hontoria@upm.es (M.H.G.); ignacio.refoyo@upm.es (I.R.R.)

**Keywords:** performance, isolation, mood, emotional intelligence, resilience, quarantine, recovery, RPE

## Abstract

The spread of COVID-19 has altered sport in Spain, forcing athletes to train at home. The objectives of the study were: (i) to compare training and recovery conditions before and during the isolation period in handball players according to gender and competitive level, and (ii) to analyse the impact of psychological factors during the isolation period. A total of 187 participants (66 women and 121 men) answered a Google Forms questionnaire about demographics, training, moods, emotional intelligence, and resilience sent using the snowball sampling technique. *T*-test and analysis of variance (ANOVA) were used to compare sport level and gender differences. Linear regressions were used to analyse the psychological influence on training. Handball players reduced training intensity (in the whole sample; *p* = 0.44), training volume (especially in professional female handball players; *p* < 0.001), and sleep quality (especially in professional male handball players; *p* = 0.21) and increased sleep hours (especially in non-professional female players; *p* = 0.006) during the isolation period. Furthermore, psychological factors affected all evaluated training and recovery conditions during the quarantine, except for sleep quantity. Mood, emotional intelligence, and resilience have an influence on physical activity levels and recovery conditions. In addition, training components were modified under isolation conditions at *p* < 0.001. We conclude that the COVID-19 isolation period caused reductions in training volume and intensity and decreased sleep quality. Furthermore, psychological components have a significant impact on training and recovery conditions.

## 1. Introduction

Currently, a virus called Coronavirus 2 (SARS-CoV-2 or COVID-19) has quickly spread to many countries around the world, causing an unexpected pandemic [1]. Consequently, quarantine and isolation periods have been imposed on citizens. In Spain, the lockdown started on 14 March, 2020 [2]. On the sporting fields, official competitions and trainings were postponed or suspended [3]. Specifically, in handball, the last matches in Spain were played on 7–8 March, 2020, and all handball players had to remain in their respective houses at least until 4 May, 2020 (almost eight weeks).

During detraining periods (off-season), monitoring the external load of players (volume and intensity) is necessary to ensure the maintenance of fitness [4]. Accordingly, technological devices such as heart rate monitors or global positioning systems (GPS) can be used to quantify the training load jointly with individual questionnaires associated with the rating-of-perceived-exertion (RPE) scale [5]. Drastic reductions in physical activity levels or cessation of training entail a decrease in the athlete’s physiological and neuromuscular adaptations, and even increases the injury risk [1]. After a period of more than 3–4 weeks, this can result in negative effects on body composition, aerobic capacity (VO_2_max), repeated sprinting ability, and strength and power in the lower limbs of the body [6]. Specifically, the physical activity reduction caused by COVID-19 (eight weeks) could decrease strength by 8–12% and fast fibre areas by 15% as well as lower muscle electrical activity [7]. Furthermore, other factors relevant to sport performance such as sleep quantity and quality, food/nutrition, hydration or mood could be affected by detraining periods [8,9].

However, the effects caused by stopping training during the quarantine were different depending on the sport and the athlete’s profile [10]. Specifically in handball, the most important differences caused by a period of non-competition (off-season) were identified in jumping performance levels, shooting velocity, maximum concentric strength of the upper limbs, and the power of upper and lower limbs [11,12]. Moreover, isolation periods associated with a lack of training sessions and official competitions in a team sport might also have led to decreased communication between players and coaching staff, and to inadequate individual training conditions [13]. Hence, isolation periods might have led players to a partial or total reversal of the adaptations produced by the training process or ‘detraining’ [7].

Together, isolation and sporting inactivity periods tend to produce psychological disorders [14]. Factors like quarantine duration, fears of infection, frustration, boredom, inadequate supplies, and inadequate information could be the most important ‘stressors’ leading to adverse mental effects [15]. However, the consequences derived from an isolation period do not have the same prevalence throughout the population. Thus, young female students were the social group that suffered the most stress, anxiety, and depression during the COVID-19 quarantine [16].

Interestingly, physical exercise positively influenced mood [17] and the athletes’ well-being [18], even in those who were in a non-specific training, recovering from injury, or off-season period [4]. Thus, the exercise load modified the subjective perception of well-being [19] and reductions in physical activity levels entailed an increase in the prevalence of the higher severity of depressive disorders [14]. In addition, variations in the strength levels registered after an off-season period in which a training programme had been implemented were related to psychological factors and self-perception [20].

Another relevant psychological area for the athlete’s performance is emotional intelligence (IE), which can be defined as the dynamic capacity to solve problems derived from the emotions of oneself and others [21]. Thus, motivation towards training and physical activity are associated with higher IE values [22]. However, this factor could be negatively affected by sport inactivity periods due to low levels of interaction between the different participants in competition (coaches, players, coaching staff, opponents, etc.) [23]. Accordingly, during isolation periods, the promotion and development of intrinsic motivation levels would become essential to decreasing anxiety and stress levels [21].

The COVID-19 quarantine has had effects at different levels (physical, physiological, psychological, emotional) due to a change in the athletes’ daily lives and training habits. The importance of this research lies in the need to know how the general state (physical, psychological, and emotional) of a handball player evolves over long periods of detraining in order to apply adequate strategies to return to physical activity. Therefore, the objectives of this study were (i) to compare training components (intensity, volume, and recovery conditions) before and during the isolation period in handball players by gender and competitive level, and (ii) to analyse the impact of psychological factors (mood, emotional intelligence, and resilience) on training components during the isolation period.

## 2. Materials and Methods

The research design corresponding to this study was non-experimental, cross-sectional, retrospective, and descriptive, based on conducting a survey through the Google Forms web platform (Google LLC, Mountain View, CA, USA).

### 2.1. Participants

The inclusion criteria were that all participants must be nationally federated handball players (Royal Spanish Handball Federation) during the 2019–2020 season. A total of 215 handball player questionnaires were collected. Surveys of injured athletes (*n* = 12), athletes not resident in Spain (*n* = 13), under 16 years old (*n* = 12), and players infected with COVID-19 during the survey (*n* = 1) were excluded from the study. There were 187 handball players in the final sample (*n* = 121; 64.7% men and *n* = 66; 35.30% women). Accordingly, the gender proportion in our study was almost identical to the Spanish handball player population (64% men vs. 36% women) [24]. The men were 23.61 ± 6.19 years old and the women were 22.65 ± 4.62 years old and had been confined for 34.66 ± 3.33 days and 34.97 ± 3.12 days, respectively. In the last two years, 26 male and 18 female players had been called up by their national or autonomic teams. Six men (5%) and one woman (1.5%) had recovered from COVID-19 when they filled out the questionnaire. Handball players were divided into two categories: professionals (‘División de Honor—ASOBAL’ and ‘División de Honor Plata’ for men and ‘Division de Honor—Guerreras Iberdrola League’ for women) and non-professionals (other national leagues). Descriptive variables of playing position, sport level, mood states, emotional intelligence, and resilience variables are shown in Table 1.

### 2.2. Instrument and Variables

The first demographic and training questionnaire was adapted to handball from football [25] jointly by five studies and sport university teachers with wide academic (more than five years) and investigative experience in the sport field. Later, the demographic questions were evaluated by two expert international handball coaches with more than 20 years’ experience and their feedback was used by the researchers to develop the definitive version of the demographic questions. Two external psychology experts assisted with adding the appropriate psychological tests for the study. The Spanish validated versions of the Profile of Mood States (POMS) [26], the Wong Law Emotional Intelligence Scale Short form (WLEIS-S) [27], and the Brief Resilience Scale (BRS-II) [28] were used to measure mood state, emotional intelligence, and resilience, respectively.

The study variables were distributed into three categories: demographic, training, and psychological variables. The demographic variables were (Q1–Q8): gender (male or female); age (years); place of residence (Spain or another country); number of days confined (days); sport level (competition category); professional or non-professional classification according to two criteria ([1] ‘Level of dedication/remuneration’: for professionals, there is a high percentage of sport contracts while non-professionals receive no remuneration for sport practice; [2] ‘Competition structure’: professionals participate in competitions structured in one group at the national level, while non-professionals compete in categories divided by groups according to geographical and economic criteria and, therefore, do not exclusively take competitive performance factors into account); selected by the national team in the last two years (yes or no); playing position (goalkeeper, wing, lateral-back, centre-back or pivot); and personal experience with COVID-19 (no experience, COVID-19 infected, or COVID-19 recovered). 

The training variables (Q9–Q18) were measured by volume (training days—‘Tdays’ and training hours ‘Thours’ per week), intensity (RPE, Likert scale 1–10), and recovery (sleep hours—‘Shours’ and sleep quality—‘Squality’, Likert scale 1–10). The psychological variables and their reliability in our study were: (1) Emotional Intelligence (Q19–Q34) [self-emotion appraisal (SEA; α = 0.81), others’ emotion appraisal (OEA; α = 0.74), use of emotion (UOE; α = 0.88), and regulation of emotion (ROE; α = 0.87)]; (2) Mood State (Q35-Q64) [tension-anxiety (α = 0.81), depression-dejection (α = 0.74), anger-hostility (α = 0.84), vigour-activity (α = 0.85), fatigue-inertia (α = 0.81), and friendliness (α = 0.79)]; and (3) Resilience (Q65–Q71) (α = 0.71). According to the previous literature, our mood variables [26], emotional intelligence [27], and resilience [28] showed excellent to acceptable reliabilities.

### 2.3. Procedures

The final version of the survey was formatted into a Google Forms Questionnaire (see Appendix A) and was sent via WhatsApp to personal contacts and published on Twitter using the snowball sampling technique [29]. Snowball sampling is a method of gathering information to access specific groups of people. The researcher asks the first few samples, who are usually selected via convenience sampling. The existing study subjects recruit future subjects among their acquaintances. Sampling continues until data saturation. This method is the most effective when the members of the population are not easily accessible [30]. One follow-up was sent to the WhatsApp contacts to improve the response rate after five days. Although we do not know the response rate, the final number of participants can be considered as a very representative dataset [31]. The questionnaire was available online for ten days starting on 16 April, 2020, just one month after the state of alarm was declared in Spain [2]. These dates were selected due to the special situation of Spain, which at that time had the second highest total number of cases of COVID-19 infections [32]. The questionnaire was open and anonymous to verify the sincerity of the answers. An unlimited time to complete the survey was provided to all athletes. Once the deadline for admitting surveys was closed, they were reviewed to remove contradictory responses (checking the congruence between the data provided by the players) or repeated (checking two or more submissions with the same responses in a short period of time), deleting one response from the database. All participants signed an informed consent form before completing the survey. This study was approved by the Ethics Committee of the Polytechnic University of Madrid.

### 2.4. Data Analysis

The data were described by arithmetic mean (*M*) and standard deviation (*SD*). The normal distribution of the variables was checked using the Kolmogorov–Smirnov and Shapiro–Wilk tests. Paired sample *t*-tests were used to compare the pre-isolation and isolation periods and independent sample *t*-tests were performed to check gender differences [25]. When statistically significant differences were found, the effect size was estimated using Cohen’s d index (d) [33], establishing two cut-off points: medium effect (0.30) and large effect (0.60). The confidence interval for the effect size was set at 95% and the percentage of change was calculated by (% = (M1 − M2/M1) × 100). The ANOVA of two factors was used to analyse the differences between professionals and non-professionals, male and female, and the interaction of both [34]. To set the differences between groups, a post-hoc analysis was carried out using the Bonferroni test. Finally, two-step hierarchical regression was performed to analyse the relationships between the psychological and training variables. IBM SPSS Statistics software (SPSS 25.0. IBM Corp., Armonk, NY, USA) was used for the mathematical calculations. The level of significance was set at *p* < 0.05.

## 3. Results

The analysis by gender showed differences in the psychological variables of depression (*t*_(185)_ = −2.54; *p* = 0.012) and fatigue (*t*_(185)_ = −2.02; *p* = 0.045) with lower values for women and higher values for men in vigour (*t*_(185)_ = 2.18; *p* = 0.03), SEA (*t*_(185)_ = 2.23; *p* = 0.027) and ROE (*t*_(185)_ = 2.14; *p* = 0.034). Moreover, a higher percentage of women in the sample played in higher categories than men *p* = 0.036.

### 3.1. Results by Isolation Period

For the whole group, RPE, Tdays, Thours, and Squality were reduced while Shours increased (all *p* < 0.001). Similar results were obtained when the analysis was carried out by gender. Only the Squality (*p* = 0.005) in men and Tdays (*p* = 0.001) and Squality (*p* = 0.001) in women presented different significant values (see Table 2).

### 3.2. Results by Gender and Sport Level

Result are summarised in Table 3. With regard to training conditions (volume and intensity), professional players had higher RPE values than non-professionals during confinement (*F*_(1,183)_ = 4.09; *p* = 0.044) and trained more days than non-professionals before (*F*_(1,183)_ = 51.58; *p* < 0.001) and during the isolation period (*F*_(1,183)_ = 20.13; *p* < 0.001). Additionally, women trained more days than men both before (*F*_(1,183)_ = 5.17; *p* = 0.024) and during the isolation period (*F*_(1,183)_ = 7.23; *p* = 0.008). Similar results were obtained with regard to the training hours, where professional players trained more hours than non-professionals before (*F*_(1,183)_ = 69.59; *p* < 0.001) and during the isolation periods (*F*_(1,183)_ = 25.27; *p* < 0.001) and women trained more hours than men both before (*F*_(1,183)_ = 11.69; *p* = 0.001) and during the isolation periods (*F*_(1,183)_ = 4.09; *p* = 0.017). Furthermore, the interaction for gender and professional level variables was significant before (*F*_(1,183)_ = 7.49; *p* = 0.007) and during the isolation period (*F*_(1,183)_ = 12.95; *p* < 0.001).

In relation to recovery (Shours and Squality), professional players slept more hours than non-professionals (*F*_(1,183)_ = 20.47; *p* < 0.001) and had better sleep quality (*F*_(1,183)_ = 7.66; *p* = 0.006) before the isolation period. Moreover, the interaction for gender and professional level variables was only significant for the sleep hours before the isolation period (*F*_(1,183)_ = 5.18; *p* = 0.017). For the rest of the comparisons, no significant effects were detected (*p* > 0.05).

### 3.3. Results by Psychological Variables

Five two-step hierarchical regression analyses were performed using RPE, Tdays, Thours, Shours, and Squality as the criterion in each case. Mood status (tension-anxiety, depression, anger, vigour, fatigue, and friendship) was entered at the first step while emotional intelligence (SEA, OEA, UOE, and ROE) and resilience were entered at the second step (see Table 4).

According to the RPE criterion, the model was non-significant at step 1 (*p* > 0.05). At step 2, UOE and resilience were significant predictors (*F*_(11,175)_ = 2.22, *p* = 0.02, *R*^2^ = 0.12, β = 0.23 and β = −0.17, respectively). The Δ*R*^2^ was significant from step 1 to step 2 (*p* = 0.015).

Based on the Tdays criterion, depression, vigour, and fatigue were significant predictors (*F*_(6,180)_ = 3.56, *p* < 0.001, *R*^2^ = 0.11, β = 0.25; β = 0.24 and β = −0.28, respectively) at step 1. At step 2, depression and fatigue retained significance and UOE was a significant positive predictor (*F*_(11,175)_ = 3.32, *p* < 0.001, *R*^2^ = 0.17, β = 0.24). The Δ*R*^2^ was significant from step 1 to step 2 (*p* = 0.018).

In relation to the Thours criterion, depression, vigour, and fatigue were significant predictors (*F*_(6,180)_ = 3.56, *p* = 0.01, *R*^2^ = 0.09, β = 0.25; β = 0.22 and β = −0.26, respectively) at step 1. At step 2, depression and fatigue retained significance and UOE was a significant positive predictor (*F*_(11,175)_ = 2.66, *p* < 0.001, *R*^2^ = 0.14, β = 0.19). The Δ*R*^2^ was not significant from step 1 to step 2 (*p* > 0.05).

Regarding the Shours criterion, the model was not significant in step 1 or step 2 (*p* > 0.05). For the Squality criterion, the model was non-significant at step 1 (*p* > 0.05), but at step 2, tension-anxiety was a significant predictor (*F*_(11,175)_ = 2.11, *p* = 0.02, *R*^2^ = 0.12, β = −0.25). The Δ*R*^2^ was not significant from step 1 to step 2 (*p* > 0.05).

## 4. Discussion

This study is one of the first to analyse the impact of the isolation period caused by COVID-19 on the training (intensity and volume) and recovery conditions (quantity and quality of sleep) of professional and non-professional handball players according to the influence of transitory psychological factors (moods) and personality trait (emotional intelligence and resilience). Based on the results yielded: (i) training and recovery conditions of the handball players were modified during the isolation period, reducing the intensity-RPE (in the whole sample), volume-Tdays and Thours (especially in professional female handball players) and sleep quality-Squality (especially in professional male handball players) and increasing sleep hours-Shours (especially in non-professional female players); and (ii) the psychological factors analysed (mood, emotional intelligence, and resilience) had an impact on training and recovery conditions, except for sleep quantity, during the Covid-19 lockdown.

### 4.1. Training Conditions

Training levels during the isolation period decreased in both intensity (RPE, *p* < 0.001, *d* = −0.93 (−54.44%)) and volume (Tdays, *p* < 0.001, *d* = −0.53 (−14.42%); Thours, *p* < 0.001; *d* = −1.28 (−89.56%)) in the whole sample. This change could activate the reversibility principle in the players, producing anatomical, functional, and physiological maladjustments [3]. Accordingly, the isolation period could cause negative effects like the reduction of cardiorespiratory capacity, deceleration of the metabolic process, a decrease in muscle activity and the process of energy generation, and a decrease in hormonal production [7]. In addition, other detrimental effects on performance such as collective tactical disorganisation or the lack of interaction between teammates could appear in team sports [13].

One possible factor for the decrease in training levels during quarantine could have been the lack of equipment or insufficient space to exercise. This fact implies that the intensity and specificity of the training could have been replaced by global exercises at a constant speed [20]. Thus, the neuromuscular adaptations of the training process achieved before the isolation period have gradually disappeared, losing the technique and speed-power in the basic skills of a team sport [12]. Furthermore, movement restrictions have caused a deterioration of the players’ fitness due to the impossibility of applying for strength training programs, producing what is known as ‘long-term atrophic decrease’ [35].

However, the training load does not seem to have decreased in the same way among all the players, especially when considering the volume (days and hours of training). Regarding the competition level, there was a greater training time reduction in professional handball players (Tdays, *p* = 0.004; Thours, *p* < 0.001) than in non-professionals (Tdays, *p* > 0.05; Thours, *p* > 0.05). Similar results were found by Skoufas et al. [36], who demonstrated that athletes with a higher competitive level reduced their training volume more than others during the off-season or non-competitive periods due to the higher initial levels of physical activity. In addition, the lack of qualified staff (coaches, physical trainers, etc.) to guide the physical-sport activity and the absence of contact and interaction with teammates could also be explanatory causes [37]. This activity reduction could lead to an increase in the body fat percentage, a decrease in the lean mass percentage, a reduction in sprint ability, a decrease in the rate of production of muscular energy, and/or a reduction in aerobic capacity [13]. 

Regarding gender, a greater reduction in training volume was observed in men (Tdays, *p* < 0.001, *d* = −0.54 (−17.64%); Thours, *p* < 0.001, *d* = −1.13 (−90.88%)) than in women (Tdays, *p* = 0.003, *d* = −0.56 (−9.39%); Thours, *p* < 0.001, *d* = −1.73 (−87.31%)). Similar results by gender were detected in the study of Giustino et al. [38] in which men reduced the amount of exercise and energy expenditure during the isolation period. Furthermore, men preferred to do physical activity outdoors more frequently than women [39]. However, when gender and competitive level were considered together, the decrease in training volume was greater in professional female players (Tdays, *p* < 0.001; Thours, *p* < 0.001) than in professional male players (Tdays, *p* = 0.021; Thours, *p* > 0.05). This result could be biased by the presence of a greater number of women who participated in the professional handball category (46.97%) compared to the number of professional male handball players (38.02%). The physical activity levels in professional female handball players before the isolation period were higher and the reduction was greater. Along this line, professional female athletes reduced their training volume more during quarantine (76%) than professional males (74%). Less communication with coaches and teammates, poorer adaptation to physical activity planning, a deterioration of rest and recovery conditions, and the thought of restarting competition in the long-term could be some of the factors that explain the difference in training volume by gender [10].

From a psychological perspective, several factors seem to have mitigated the decrease in training intensity and volume during the isolation period. The psychological component that had the greatest impact on training conditions was the use of emotions—UOE (RPE, *R*^2^ = 0.12, *p* = 0.01; Tdays, *R*^2^ = 0.17, *p* = 0.01; Thours, *R*^2^ = 0.14, *p* = 0.03). Team sport players tend to manage and use their emotions more easily and frequently for different purposes, better adapting to the environment’s conditions [21]. Therefore, a greater skill in the UOE would have enabled handball players to maintain higher external load parameters during the isolation period. Greater contact with other teammates and friends, the clear and simple design, and planning of workout routines for training and encouraging an exercise-prone mood seem to have facilitated the control and management of negative feelings interrupting, to a lesser extent, training conditions [10,17].

Interestingly, the present study yielded two surprising a priori results associated with moods and personality traits. The first connects, as opposed to other studies in different fields [22,40], a lower resilience (BRS) with a higher training intensity (*R*^2^ = 0.12, *p* = 0.046). Difficulty and confusion in measuring perceived training intensity (RPE) in isolated situations could have caused a relatively lower effort record, associated with a higher degree of control and experience [19]. The second finding links higher levels of depression to a higher training volume (Tdays, *R*^2^ = 0.17, *p* = 0.01; Thours, *R*^2^ = 0.14, *p* = 0.01;). Players seek relief from depression and anxiety symptoms in physical-sport activity, thanks to a high commitment to sport excellence and high levels of athletic identity [41].

### 4.2. Recovery Conditions

In relation to the recovery of physical activity pre- and post-isolation, the whole sample presented an increase in quantity of sleep (Shours, *p* < 0.001, *d* = 0.69 (10.20%)) and a decrease in quality of sleep (Squality, *p* < 0.001, *d* = −0.34 (−15.20%)). Considering the correlation between physical activity levels and mental well-being [18], the limitation or cessation of exercise could cause a deterioration in sleep quality, which is closely associated with well-being.

The factor with the greatest negative impact on sleep quality during the isolation period was tension-anxiety (*R*^2^ = 0.12, *p* = 0.01). Isolation conditions could lead to worries about the athlete’s fitness and restarting competition. These could be the main precursor factors of psychological disorders during quarantine such as stress, depression, or irritability [1,15]. However, other behaviours could have affected sleep quality during the isolation period such as nutritional changes and a sedentary lifestyle [10]. On one hand, a decrease in the consumption of foods rich in vitamin D and carbohydrates could have caused a worsening in the athlete’s recovery capacity [3]. On the other hand, negative changes in lifestyle (lower levels of physical activity, higher use of technologies, higher consumption of alcohol and other addictions, etc.) could have caused an alteration in sleep quality and fatigue perception [37]. 

However, tension-anxiety levels were higher in professional male handball players. Together, the need to keep an adequate physical condition (self-perception of the physical profile) [20] to compete at top levels (‘Liga ASOBAL’ and ‘Liga de Division de Honor Plata’) and the impossibility of training under appropriate conditions could have caused adverse mental health states [42] that did not allow for optimal rest conditions. Accordingly, increases in fatigue and injury risk could lead to a decrease in the player’s recovery capacity [43].

Regarding sleep quantity, an increase in sleep hours during the isolation period was detected in handball players, especially in non-professional female players. This change could be associated with changes in lifestyle caused by mobility restrictions. The time spent lying in bed increased and waking time was delayed, thus increasing sleep hours [44]. The greater daily flexibility of schedules in the young population, similar to handball players, could have been an opportunity to develop sleep habits that are more closely linked to their endogenous body rhythms, favouring a greater number of sleep hours [8].

Similar results were found in other sports. On a psychological and emotional level, Clemente-Suárez*,* et al. [45] detected, in Olympic and Paralympic athletes, a higher impact of the isolation period (psychological inflexibility) on professional players and on women due to greater experimentation with stressful situations. Considering the training conditions in team sports, Mon-López et al. [25] found how the frequency, duration, and intensity of training were reduced during the quarantine in football players in Spain. In relation to rest conditions, Pillay et al. [10] demonstrated the alteration of sleep patterns (reduction in sleep quality), which led to an increase in the level of fatigue and the rate of injuries in a study with 692 athletes from 15 different sports. Therefore, it seems that the effects of isolation period have generally been perceived as negative by the athletes.

### 4.3. Practical Applications

Due to the deterioration of the fitness in handball players caused by the quarantine, coaches, physical trainers, technical staff, sports institutions, and national and international sports federations should plan strategies and training programmes aimed at reducing the detraining effects according to the athlete’s gender and competitive level to avoid potential adverse effects such as injuries. However, even with continuous and accurate player monitoring, there could be a certain degree of non-control of the variables analysed due to the implications of government laws connected to public health. Thus, the impact of these variables on the player’s fitness could undergo modifications external to sport training.

Based on the scientific evidence, specific exercises should be proposed to personalise the training and recovery process of athletes, especially in handball players. Setting clear training objectives through simple training tools and resources; personalised definition of the training external load variables according to a holistic vision of the context and the sport experience of the player during the previous competitive period (injuries, minutes played, competitive experience, etc.) [13]; organise a ‘player support network’ by the experts (coach, doctor, psychologist, nutritionist, etc.) through the use of technology (phone calls, video calls, email, etc.) to guarantee suitable physical and psychological levels for the return to competition; design an individualised home fitness training programme according to available space and equipment resources tailored to the athlete’s characteristics and current needs [3]; provide adequate recovery and rest methods (sleep and relaxation techniques, stretching, supplementation, etc.) [9]; and daily monitoring of the athlete’s well-being, physical state, recovery capacity and psychological state [19].

### 4.4. Limitations

Although this is one of the first studies on the effects of COVID-19 on handball players, some limitations should be mentioned. The unprecedent social and sport context in which the research was carried out and the isolation situation were novel regarding the sport training process and additional data would be necessary. Moreover, the final sample of the study was 187 players, which limited the statistical power and possibly resulted in a self-selection bias associated with the athlete’s gender and level of competition. Accordingly, the results should be considered with caution, especially due to the sample imbalance with a greater presence of professional female handball players. Another important aspect was that to ensure the sincerity of the answers, the questionnaires were anonymous, which implied the impossibility of confirming the athletes’ identities.

### 4.5. Future Research Lines

Future study designs could consider including more variables in relation to demographic characteristics (level of studies, place of residence, etc.), training and recovery conditions (available space, training machines, etc.), and mood (motivations, etc.) of the players. On the other hand, an improvement in the monitoring systems for the training quantity and quality would be desirable in order to draw more precise conclusions. Furthermore, conducting a longitudinal study covering the pre-, during and post-isolation periods through various measurements could provide information on how such a long detraining period influences the habits of handball players.

## 5. Conclusions

The COVID-19 isolation period had significant adverse effects on the training and recovery conditions of handball players, leading to physical deconditioning and worsening sleep conditions. Relevant training reductions in volume and intensity were detected, especially in women and professionals, while a decrease in sleep quality was identified in professional handball players, especially in men.

The psychological components had a significant impact on training and recovery conditions during the isolation period. Psychological traits associated with personality such as resilience or emotional intelligence (use of emotions—UOE) were modifying factors of the training intensity and volume, and moods, based on components such as fatigue, depression, and tension-anxiety had a greater impact on the rest and recovery conditions of the players as well as on the external load of training.

The set model of mood and personality traits (emotional intelligence and resilience) was explanatory of the training and recovery conditions of handball players during the isolation period, especially on the physical activity levels associated with a reduction in the days and hours that players use for exercise.

## Figures and Tables

**Table 1 ijerph-17-06471-t001:** Descriptive distribution of the handball players by gender (women, men, and total) according to mood state, emotional intelligence (EI) and resilience, sport level (condition and category), and playing position.

	**Mood State**												**Emotional Intelligence-Resilience**									
	**T-A**		**Depression**		**Anger**		**Vigour**		**Fatigue**		**Friendship**		**SEA**		**OEA**		**UOE**		**ROE**		**BRS**	
	***M***	***SD***	***M***	***SD***	***M***	***SD***	***M***	***SD***	***M***	***SD***	***M***	***SD***	***M***	***SD***	***M***	***SD***	***M***	***SD***	***M***	***SD***	***M***	***SD***
Women	9.65	4.25	8.26	3.64	7.94	4.19	10.74	3.86	8.65	3.95	14.61	2.89	5.04	0.81	5.30	0.91	5.09	1.21	4.51	1.17	17.21	3.97
Men	9.80	4.34	6.75	4.00	7.25	4.07	11.98	3.64	7.50	3.58	14.07	3.38	5.37	1.04	5.19	0.90	5.22	1.23	4.89	1.18	18.00	3.63
Total	9.75	4.30	7.28	3.93	7.49	4.12	11.55	3.75	7.91	3.74	14.26	3.22	5.26	0.98	5.23	0.90	5.17	1.22	4.76	1.19	17.72	3.76
	**Sport Level**												**Playing Position**									
	**Prof**		**N-Prof**		**1st Category**		**2nd Category**		**3rd Category**		**4th Category**		**Goalkeeper**		**Wing**		**Centre-back**		**Lateral-back**		**Pivot**	
	***n***	***%***	***n***	***%***	***n***	***%***	***n***	***%***	***n***	***%***	***n***	***%***	***n***	***%***	***n***	***%***	***n***	***%***	***n***	***%***	***n***	***%***
Women	31	47.00	35	53.00	31	33.30	22	47.00	13	19.70	0	0	9	13.60	19	28.80	13	19.70	14	21.20	11	16.70
Men	46	38.00	75	62.00	10	8.26	36	29.75	22	18.18	53	43.80	20	16.50	25	20.70	26	21.50	25	20.70	25	20.70
Total	77	41.20	110	58.80	41	21.93	58	31.01	35	18.71	53	28.34	29	15.51	44	23.53	39	20.86	39	20.86	36	19.25

Notes: T-A = tension-anxiety; SEA = self-emotion appraisal; OEA = other’s emotion appraisal; UOE = use of emotion; ROE = regulation of emotion; BRS = brief resilience scale; Prof = professional players; N-Prof = non-professional players.

**Table 2 ijerph-17-06471-t002:** Differences on training variables between pre-isolation and isolation period by gender.

		Pre-Isolation		Isolation Period		*R*	*p*	Cohen’s *d*		IC 95%		Change (%)
		*M*	*SD*	*M*	*SD*			*D*	*D Pooled*	*LL*	*UL*	
All	RPE	6.44	2.74	4.17	2.47	0.60	<0.001	−0.93	−0.97	−0.7	−1.1	−54.44
	Tdays	4.84	1.15	4.23	1.69	0.50	<0.001	−0.53	−0.42	−0.3	−0.7	−14.42
	Thours	9.99	3.80	5.27	3.26	0.53	<0.001	−1.28	−1.38	−1.1	−1.5	−89.56
	Shours	7.22	0.88	8.04	1.29	0.09	<0.001	0.69	0.55	0.48	0.9	10.20
	Squality	6.29	2.40	5.46	2.60	0.49	<0.001	−0.34	−0.33	−0.1	−0.6	−15.20
Men	RPE	6.41	2.62	4.07	2.58	0.59	<0.001	−0.99	−0.99	−0.72	−1.25	−57.20
	Tdays	4.69	1.20	3.98	1.69	0.39	<0.001	−0.54	−0.44	−0.28	−0.80	−17.64
	Thours	9.34	3.63	4.89	3.15	0.41	<0.001	−1.13	−1.2	−0.86	−1.40	−90.88
	Shours	7.26	0.85	8.04	1.34	0.17	<0.001	0.71	0.54	0.45	0.97	9.66
	Squality	6.21	2.45	5.57	2.61	0.53	<0.005	−0.27	−0.26	−0.02	−0.52	−11.43
Women	RPE	6.50	2.97	4.33	2.28	0.65	<0.001	−0.87	−0.98	−0.52	−1.23	−50.00
	Tdays	5.12	1.00	4.68	1.61	0.69	<0.003	−0.56	−0.42	−0.21	−0.91	−9.39
	Thours	11.18	3.83	5.97	3.36	0.69	<0.001	−1.73	−1.84	−1.33	−2.13	−87.31
	Shours	7.17	0.94	8.05	1.21	−0.06	<0.001	0.64	0.55	0.30	0.99	10.92
	Squality	6.44	2.32	5.27	2.62	0.42	<0.001	−0.47	−0.44	−0.12	−0.81	−22.13

Notes: IC = interval confidence; LL/UP = lower/upper limit; RPE = rated perceived perception; Tdays = training days; Thours = training hours; Shours = sleep quantity; Squality = sleep quality.

**Table 3 ijerph-17-06471-t003:** Differences by gender and professional sport level before and during isolation periods.

		All		*p*	All		*p*	Prof		*p*	N-Prof		*p*	Men		*p*	Women		*p*
		Prof	N-Prof		Men	Women		Men	Women		Men	Women		Prof	N-Prof		Prof	N-Prof	
Pre isolation period	RPE	6.82	6.16	0.123	6.46	6.53	0.870	6.67	6.97	0.646	6.24	6.09	0.784	6.67	6.24	0.400	6.97	6.09	0.194
	Tdays	5.53	4.43	<0.001	4.81	5.16	0.024	5.30	5.77	0.045	4.31	4.54	0.251	5.30	4.31	<0.001	5.77	4.54	<0.001
	Thours	12.55	8.46	<0.001	9.67	11.35	0.001	11.04	14.07	<0.001	8.29	8.63	0.606	11.04	8.29	<0.001	14.07	8.63	<0.001
	Shours	7.54	6.95	<0.001	7.30	7.19	0.428	7.44	7.65	0.283	7.16	6.74	0.016	7.44	7.16	0.082	7.65	6.74	<0.001
	Squality	6.91	5.89	0.006	6.33	6.47	0.705	6.85	6.97	0.828	5.81	5.97	0.745	6.85	5.81	0.021	6.97	5.97	0.090
Isolation period	RPE	4.64	3.87	0.044	4.15	4.36	0.586	4.48	4.81	0.567	3.83	3.91	0.862	4.48	3.83	0.159	4.81	3.91	0.144
	Tdays	4.95	3.85	<0.001	4.07	4.73	0.008	4.41	5.48	0.004	3.72	3.97	0.440	4.41	3.72	0.021	5.48	3.97	<0.001
	Thours	6.70	4.36	<0.001	4.97	6.09	0.017	5.30	8.10	<0.001	4.64	4.09	0.368	5.30	4.64	0.239	8.10	4.09	<0.001
	Shours	8.16	7.97	0.333	8.08	8.05	0.853	8.26	8.07	0.516	7.91	8.03	0.647	8.26	7.91	0.147	8.07	8.03	0.911
	Squality	5.53	5.35	0.661	5.59	5.28	0.436	5.70	5.36	0.577	5.49	5.20	0.586	5.70	5.49	0.681	5.36	5.20	0.811

Notes: Prof = professional players; N-Prof = non-professional players; RPE = rated perceived perception; Tdays = training days; Thours = training hours; Shours = sleep quantity; Squality = sleep quality.

**Table 4 ijerph-17-06471-t004:** Hierarchical regressions of training variables onto the psychological factors (moods, emotional intelligence, and resilience) of handball players.

Model	Predictor	RPE			Tdays			Thours			Shours			Squality		
		*β*	*t*	*p*	*β*	*t*	*p*	*β*	*t*	*p*	*β*	*t*	*p*	*β*	*t*	*p*
Step 1	Tension-Anxiety	−0.12	−1.30	0.195	−0.08	−0.91	0.362	−0.17	−1.92	0.056	0.11	1.23	0.222	−0.23	−2.62	0.01
	Depression	0.18	1.59	0.113	0.25	2.34	0.02	0.25	2.32	0.02	−0.03	−0.23	0.817	0.06	0.51	0.614
	Anger	0.06	0.54	0.589	0.16	1.48	0.14	0.15	1.38	0.169	−0.07	−0.60	0.552	0.06	0.54	0.589
	Vigour	0.26	2.69	0.008	0.24	2.49	0.01	0.22	2.27	0.02	0.03	0.29	0.771	0.17	1.71	0.09
	Fatigue	−0.03	−0.35	0.728	−0.28	−3.03	<0.001	−0.26	−2.72	0.01	−0.01	−0.12	0.906	−0.12	−1.22	0.225
	Friendship	−0.04	−0.40	0.692	−0.02	−0.22	0.824	−0.01	−0.08	0.939	−0.19	−2.06	0.04	−0.14	−1.52	0.129
	*F*/ *R*^2^/ Adj. *R*^2^	1.51/0.05/0.02		0.176	3.56/0.11/0.08		<0.001	3.05/0.09/0.06		0.01	1.07/0.04/0.00		0.381	2.47/0.08/0.05		0.026
Step 2	Tension-Anxiety	−0.13	−1.49	0.138	−0.09	−1.05	0.294	−0.17	−1.87	0.063	0.10	1.05	0.293	−0.25	−2.76	0.01
	Depression	0.18	1.56	0.122	0.27	2.48	0.01	0.30	2.74	0.01	−0.03	−0.29	0.776	0.11	0.95	0.344
	Anger	0.01	0.05	0.957	0.12	1.09	0.279	0.14	1.24	0.215	−0.12	−1.03	0.306	0.00	0.04	0.971
	Vigour	0.19	1.85	0.066	0.15	1.58	0.117	0.14	1.42	0.158	0.00	0.00	0.998	0.12	1.20	0.234
	Fatigue	−0.03	−0.36	0.723	−0.28	−3.06	<0.001	−0.26	−2.80	0.01	−0.01	−0.05	0.958	−0.14	−1.45	0.148
	Friendship	−0.08	−0.88	0.379	−0.06	−0.65	0.52	0.01	0.05	0.962	−0.22	−2.26	0.03	−0.08	−0.79	0.43
	SEA	0.11	1.21	0.228	0.10	1.20	0.231	0.09	1.05	0.294	−0.02	−0.20	0.841	0.14	1.59	0.113
	OEA	0.06	0.76	0.45	0.08	1.03	0.305	0.01	0.18	0.858	0.09	1.10	0.274	−0.11	−1.37	0.173
	UOE	0.23	2.59	0.01	0.24	2.74	0.01	0.19	2.20	0.03	0.15	1.66	0.099	0.15	1.67	0.097
	ROE	−0.08	−0.88	0.378	−0.09	−0.98	0.331	−0.06	−0.66	0.511	−0.15	−1.53	0.129	−0.19	−1.95	0.053
	BRS	−0.17	−2.01	0.046	−0.08	−0.93	0.352	0.07	0.79	0.433	−0.01	−0.08	0.939	0.00	−0.05	0.959
	*F*/ *R*^2^/ Adj. *R*^2^	2.2/0.12/0.07		0.017	3.32/0.17/0.12		<0.001	2.66/0.14/0.09		<0.001	1.15/0.07/0.01		0.322	2.11/0.12/0.06		0.02
	Δ F/Δ *R*^2^	2.92/0.07		0.015	2.83/0.07		0.018	2.07/0.05		0.071	1.24/0.03		0.289	1.64/0.04		0.152

Notes: RPE = rated perceived perception; Tdays = training days; Thours = training hours; Shours = sleep quantity; Squality = sleep quality; SEA = self-emotion appraisal; OEA = other’s emotion appraisal; UOE = use of emotion; ROE = regulation of emotion; BRS = brief resilience scale.

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
