# Peer review of "The Impact of Covid-19 and the Effect of Psychological Factors on Training Conditions of Handball Players"

_ijerph, 2020, doi:10.3390/ijerph17186471_

Round 1
Reviewer 1 Report
Thank you very much for providing this interesting paper. Very nice work has been done here. Overall, I think the paper is well written and clear for most parts. A few minor changes indicated in the feedback file might help you to improve your outcome even more.

Reviewer 2 Report
The authors have provided a systematic description of psycho-social factors impacted by the changes in training quantity and quality brought about by COVID-19 Public Health Orders initiated by the Spanish Health authorities. The investigation and methods are well described. In the discussion and practical applications section to the manuscript, the authors should consider a statement that reflects the lack of control from a sporting perspective on some of the impacting factors. The fact that the situation was governed by public health orders and thus not controllable by the sports is important, even though the authors have done well in justifying reflective link to sport practices involved in understanding de-training and off-season adaptive processes.
Ln 128-136; Please check the line spacing in the manuscript to ensure consistency
Ln 341; Should be 'greater' not 'great'
Ln 355; Suggest changing 'baggage' to 'experience'
Table 3; Although I recognize why the authors have included the columns for Men and Women with the sub columns for Prof and Non-Prof is to improve the readers ability to directly compare the data, this is a duplication of data already presented in the table and as such is unnecessary and should be deleted.
Reviewer 3 Report
Overall Comments:
The authors present an interesting study that addresses a current issue and, according to expectations, a situation to take into account with professional athletes in the near future. However, it is not explicitly stated what type of research they propose, what is its design or the desired purpose. A further explanation of the motivations of the study that led to the stated objectives would also be desirable.
The instrument to obtain the information (questionnaire) has not been previously validated, it would have been desirable to carry out a pilot study. Nor is the elaboration protocol followed by the experts described, which could compensate for the non-validation of the questionnaire.
It would be convenient to provide the identification number of acceptance by the ethics committee of the Polytechnic University of Madrid.
The statistical treatment of the results is very complete and accurate, highlighting the comprehensiveness proposed. Likewise, the results presented are significant and relevant. A comparison with the results obtained in similar studies in other individual or team sports would have been desirable.
The conclusions are firmly based on the results obtained and allow to establish valuable practical applications in the event of future confinements.
The study has merit and scientific interest. My recommendation is “accepted with modifications”.
Specific Comment:
Nowadays, there is no scientific evidence of consensus on the origin of the pandemic, so it is not appropriate to make statements about its origin, especially when it is not an important aspect of the research (lines 32-33).
The inclusion criteria indicated is to be "members of the Royal Spanish Handball Federation" and play in a national league. It would be clearer simply to indicate as a criterion that they must be "federated national handball players" (lines 89-90).
It should be understood that the people responsible for the questionnaire, in addition to university professors, are "researchers", this condition being the relevant one in the study (lines 107-108).
The questionnaire was open and anonymous and also was available for a period of ten days to be completed (lines 142-146). Did you use any procedure to avoid sending multiple responses by the same subject? Which one?
Why were demographic variables such as place of residence or level of studies not taken into account? (lines 117-127) or training variables such as available space or equipment? (lines 128-130). Consider using them for future researches.
Finally, it would be desirable to include a copy of the questionnaire as an annex or a link to the online questionnaire.
